# Real-time Smartphone Activity Classification Using Inertial Sensors—Recognition of Scrolling, Typing, and Watching Videos While Sitting or Walking

**DOI:** 10.3390/s20030655

**Published:** 2020-01-24

**Authors:** Sijie Zhuo, Lucas Sherlock, Gillian Dobbie, Yun Sing Koh, Giovanni Russello, Danielle Lottridge

**Affiliations:** 1Department of Electrical, Computer and Software Engineering, University of Auckland, Auckland 1010, New Zealand; szhu842@aucklanduni.ac.nz (S.Z.); lshe485@aucklanduni.ac.nz (L.S.); 2School of Computer Science, University of Auckland, Auckland 1010, New Zealand; g.dobbie@auckland.ac.nz (G.D.); ykoh@cs.auckland.ac.nz (Y.S.K.); g.russello@auckland.ac.nz (G.R.)

**Keywords:** smartphone activity recognition, smartphone IMU sensors, real-time classification, machine learning

## Abstract

By developing awareness of smartphone activities that the user is performing on their smartphone, such as scrolling feeds, typing and watching videos, we can develop application features that are beneficial to the users, such as personalization. It is currently not possible to access real-time smartphone activities directly, due to standard smartphone privileges and if internal movement sensors can detect them, there may be implications for access policies. Our research seeks to understand whether the sensor data from existing smartphone inertial measurement unit (IMU) sensors (triaxial accelerometers, gyroscopes and magnetometers) can be used to classify typical human smartphone activities. We designed and conducted a study with human participants which uses an Android app to collect motion data during scrolling, typing and watching videos, while walking or seated and the baseline of smartphone non-use, while sitting and walking. We then trained a machine learning (ML) model to perform real-time activity recognition of those eight states. We investigated various algorithms and parameters for the best accuracy. Our optimal solution achieved an accuracy of 78.6% with the Extremely Randomized Trees algorithm, data sampled at 50 Hz and 5-s windows. We conclude by discussing the viability of using IMU sensors to recognize common smartphone activities.

## 1. Introduction

Human activity tracking using smartphone sensors has been a key concern since smartphones have been released, in part due to smartphones’ physical size, ubiquity, unobtrusiveness and ease of use. Whole-body human activities such as walking and running have been investigated but physically smaller human activities such as reading or typing have not been explored. Knowing whether a user is scrolling, typing or watching videos is valuable in our endeavors to better understand users. Digital content and experience providers benefit from tracking users’ smartphone activities to inform personalization, as demonstrated by several recent patents [1,2,3]. For example, news and social media applications log when users watch videos or read in order to present personalized content which is more likely to be engaging to each user [4]. If a news or social media app uses inertial measurement unit (IMU) sensors for this, they can present the user with more personalized content (i.e., more videos or more text, depending on what the user uses and likes).

The other side of this investigation is a question of security. Currently it is not possible to access user action logs directly, due to smartphone data access privileges. Detailed logs of all user smartphone activities may be considered private or sensitive. As of 2019, smartphones do not ask users for permission to access IMU sensor data. A phone would need to be “rooted,” in other words, an application would need to gain root access control, in order to access activity data directly. This may be because current IMU applications do not seem as privacy-sensitive as other permission-protected sensors such as Global Positioning System (GPS), the microphone and camera. However there is growing evidence that IMU sensors can be used for nefarious purposes, such as location tracking [5] and password sniffing [6]. In response to this trend, new privacy applications are being introduced that enforce runtime control policies to protect IMU sensors from misuse [7]. To inform whether IMU sensors should be permission-protected, we aim to understand the precision with which our human smartphone activities can be estimated from smartphone standard internal sensors including a accelerometer, gyroscope and magnetometer. This investigation fits into an emerging literature on human activity recognition with IMU sensors [8,9,10,11,12,13,14,15,16,17].

Over the last decade, smartphone IMU sensors have been used to classify general user behaviors including sitting, walking, jogging, running or walking up and down stairs [8,9,10,11,12,13,14,15,16,17]. Typically, such research first collects raw sensor data, extracts time domain or frequency domain features from the raw data, and then uses the extracted features to train a machine learning model to classify the specified user activities. Using these methods, researchers have developed machine learning models which can classify user activities with greater than 95% accuracy [8,9,12,14,15]. The area of human activity recognition has focused on recognizing physically larger human activities (walking, etc.) [8,9,10,11,12,13,14,15,16,17] but to date none have tried to recognize what the person is doing on the smartphone itself, namely reading, typing, scrolling or watching. A user’s activity on their smartphone can be estimated to some degree through knowledge of the app they are using. For instance, in a news application, does the user tend to read articles or watch news programs? There are privacy concerns associated with monitoring users’ application use, making this approach infeasible for use on a large scale. A person who is typing or scrolling on their phone still produces movements and vibrations which can be picked up by the phone’s inertial sensors, similar to when a person is walking or running. Figure 1 shows a comparison of a 10 s triaxial accelerometer reading between typing and watching while seated. Two observable differences are the swap of x and y axes between these two activities and the greater variation when the participant is typing. These movements may be small but they can still be used in the same process as outlined above to build a model and predict the user’s activity, though we currently do not know what precision is achievable.

In this study, we used smartphone IMU sensors to classify the following user smartphone activities: scrolling, typing, watching and non-use, while seated or walking. We conducted a study with human participants, using an Android app which we developed to collect labelled IMU sensor data while they were carrying out the activities which we aimed to classify. The app led the user to perform the activities for two-minute blocks, with an optional rest between blocks. We performed a machine learning-based method for smartphone activity recognition. This method achieved an accuracy of 78.6% in classifying eight types of activities: scrolling, typing, watching and non-use, while sitting and walking. The contributions of this paper are:the first investigation of recognition of human smartphone activities using IMU sensors,a novel machine learning model that classifies smartphone activities, suitable features and time window sizes, andthe accuracy that the model and parameters can achieve, along with a discussion of feasibility and implications.

These contributions will be useful for understanding the precision of smartphone activity tracking with IMU sensors for purposes such as personalization and when considering whether IMU sensors present security concerns due to their ability to recognize private behavior.

The remainder of the paper is organized as follows. Section 2 presents recent studies on human activity recognition. The method we used to collect data and the study setup are presented in Section 3. The results of the study and discussion about the findings are presented in Section 4 and Section 5, followed by the conclusion and future work in Section 6.

## 2. Related Work

The study of modern human activity recognition began from analysis of visual data from pictures and film [18] and shifted into using movement-based wearable sensing devices. Both of these methods used specialized technology which is not readily available to a typical consumer and is difficult for them to use in their everyday life. For these reasons, the leading platform for activity recognition has shifted to the smartphone [19,20]. The rich set of sensors integrated into smartphones, such as an accelerometer and gyroscope, make the smartphone capable of collecting data similar to that obtained from a wearable sensing device. Smartphones have the added benefit of being relatively ubiquitous and accessible. Their powerful data transmission technologies, Wi-Fi and Bluetooth, make them a practical platform for personal activity recognition.

However there are limitations to the use of smartphones for activity recognition. Continuous sensing is a challenge [21] as this can take a significant toll on the phone’s battery life [22] as well as generate more data than can be stored feasibly on the phone. Gathering data based on a sampling frequency can help to conserve battery and adaptive algorithms are being developed to adjust this frequency based on the activity which is occurring [21]. Running the classification algorithm on the phone requires more processing power than is readily available [22]. At present outsourcing computation to a more powerful machine is currently the best approach with current smartphone technology, although it introduces time delays. Outsourcing is the method we have used in our research.

Most studies which use smartphones for activity recognition use the accelerometer as a part of their input data [8,9,10,12,13,17]. The smartphone accelerometer has also been extensively utilized in research fields adjacent to human activity recognition. For instance, smartphone sensors have been used to estimate the user’s mood [23] and stress level [24]. Another study found that smartphone accelerometer can be used to counterbalance the movement of the phone while walking and thus improve typing efficiency and reduce the error rate of the user [25]. Smartphone accelerometers can also be used to detect changes in the surrounding environment; one study shows the possibility of using the accelerometer to detect the vibration of the nearby keyboards and thus decode what is being typed [26]. Some studies have suggested that the use of the gyroscope, together with the accelerometer, can improve classification accuracy [8,9,27]. With these existing applications in mind, we investigate whether internal smartphone sensors are reliable enough to produce an accurate result for typical smartphone activities such as scrolling feeds, typing and watching videos.

Table 1 provides an overview of the sensors and features used in research on human activity recognition with smartphones. Accelerometers and gyroscopes are the most common sensors used. Note that one study did not perform feature extraction because the machine learning model used was a deep convolutional neural network, and the layers in the network acted as a hierarchical feature extractor [9].

Many machine learning solutions have been employed in human activity recognition. These machine learning models range from traditional models such as k-nearest neighbors (KNN) [11,12,13,14,15,16] to neural networks [8,9,10,11], and from supervised learning models [8,9,10,11,12,13,14,15,16] to the unsupervised model [17]. In this prior work, the KNN solution proposed by Ustev [12] outperforms other solutions with the highest accuracy of 97%. However, this does not mean that the KNN classifier is the best solution for all situations because there were different setups in the different studies. Recently, a model which combines both supervised and unsupervised methods to classify human activities has become popular; Lu et al. show that a hybrid solution can outperform the models which use a supervised method only [17]. The MCODE method proposed by Lu et al. outperforms the following classifiers—GMM, HC, K-means++, K-medoids and SC.

A 2015 survey which reviewed 30 studies on online classification of physical activities with smartphones concluded that the commonly used classifiers are relatively simple compared to the classifiers which operate on a desktop computer [28]. This simplicity is due to smartphones’ limited resources and computational power.

Several studies looked at human smartphone gestures for the purpose of continuous authentication. Sitova et al. investigated if they were able to authenticate users based on patterns in how users hold, grasp, and tap their phones while sitting and walking and achieved a relatively low error rate of 7.10% [29]. Kumar, Phoha, and Serwadda fused swiping gestures, typing patterns and phone movement patterns for continuous authentication and achieved 93.33% accuracy [30]. There are no prior studies on recognizing typical smartphone activities for the purpose of classifying smartphone behaviors.

In our research, we investigate what accuracy can be achieved in classifying typical smartphone activities (scrolling, typing, watching and non-uses, while sitting or walking) using internal sensor data along with simple algorithms.

## 3. Methods

We conducted a study where human participants completed specified activities on smartphones and where the data was used for real-time classification. We first discuss an overview of this work and then we describe the study setup and software used.

We follow a general approach to classifying human activities using IMU sensors. First, we collect raw data from IMU sensors. Next, we extract time-domain features (e.g., mean, standard deviation) and frequency-domain features (e.g., energy) from the raw data. The features extracted are then segmented and used as the input of a machine learning model. The model must be trained with sufficient data. Then the model is used for real-time classification. The accuracy of the model is validated by separating the data into two disjoint sets, where one set is used as the training set, and the other is used as the test set. The accuracy of the model is determined by how correctly the test set is predicted using the training set, which is referred to as cross-validation. This process is summarized in Figure 2.

### 3.1. Study Setup

In this study, participants were asked to perform a series of tasks using a test phone (Samsung Galaxy S7) under the conditions of sitting and walking (Figure 3). Another part of the study focused on cognitive load, which is reported on elsewhere [31,32]. For each condition of sitting and walking, there were four tasks: reading an article, scrolling through a social media feed, typing sentences, and watching short videos (Figure 4).

The content of the activities was chosen to be of general interest without violence or explicit content. The articles for the reading activities were selected from news published in the NZ Herald (The New Zealand Herald is a daily newspaper published in Auckland, New Zealand. https://www.nzherald.co.nz/) or on Medium (Medium is an online publishing platform. https://medium.com/), and they had a suggested reading time of 2 to 3 min. The participants were asked to answer one multi-choice question for each article after they had read it. The purpose of the question is to motivate the participants to read the articles with enough care that they can answer basic questions about the content. The sentences selected for the typing activity were chosen from a dataset used for evaluation of text entry techniques [33]. The phrases provided by the study are usually moderate in length, easy to remember, and representative of the target language. During the typing activity, the participants were asked to type the phrases as they appeared on screen. The order of the phrases was random, and “Qwerty” keyboard was used. The social media feeds were taken from Reddit (Reddit is a social news aggregation, web content rating, and discussion website. https://www.reddit.com/), sourced by taking long screenshots using a smartphone, and then the images were inserted into our application. The three videos selected for the application were downloaded from YouTube (YouTube is a video-sharing platform. https://www.youtube.com). These were a movie trailer, a funny video and an educational video. The selection of content aimed to cover the general types of activities which people may perform in their daily life.

The sitting and walking conditions had an additional activity, namely non-use, where baseline data was collected. For these non-use tasks, participants were asked to place the smartphone anywhere they preferred while sitting or walking.

Each activity was two minutes long. Short breaks could be taken between activities. Participants were asked to perform a total of 10 tasks. The order of conditions and tasks (excluding the baseline tasks) was randomly generated to minimize any bias that could be introduced by the sequence of tasks. The baseline task would always appear at the beginning of the sequence of tasks for the corresponding condition.

After the tasks, participants were asked to fill out a questionnaire which included demographic information and typical phone usage. We collected age, gender, dominant hand, phone model, typical smartphone keyboard layout, and usual phone usage. The questions asked how often the participant would typically engage in the activities tested in the study: that is, the frequency of scrolling feeds, reading articles on their phone, typing and watching video, while sitting and walking. The questionnaire data was used to investigate relationships between the participants’ daily reported usage and collected data.

Two software applications were developed for this research: an Android app which participants interacted with during the study and which collects smartphone IMU sensor data (described above), and a python application which runs on a desktop computer to process the data and provide real-time classification. The python application processed the data collected from the smartphone and performed feature extraction and classification. The application also performed several validation functions including 5-fold cross-validation. It generated correlation tables, confusion matrices and F1-scores.

#### Participant Recruitment and Data Acquisition

We recruited 21 participants for our study. Participants were recruited from the software engineering and computer science students who were studying at the University of Auckland. All participants gave their informed consent for inclusion before they participated in the study. The study was conducted in accordance with the Declaration of Helsinki, and the protocol was approved by the University of Auckland Human Participants Ethics Committee (UAHPEC) with reference 023008. Among these participants, 15 were male (71.4%), and the remainder were female (28.6%). The age of the participants ranged from 21 to 42 years, with a mean age of 25.1 and a median of 23. Inclusion in the research required that the participants be familiar with using a smartphone and have no physical limitations preventing them from interacting with the device. To maintain the validity of our results and consistency between participants, a study protocol was developed and strictly adhered to.

The sensor data was transmitted from the smartphone to computer via Bluetooth with a frequency of 10 Hz. Transmission via Bluetooth was selected over Wi-Fi due to its availability, low cost and ease of use. As the sensor data was sent, it was also labelled as the activity that was currently taking place in the study; for example, while a participant was engaged with the scrolling task, we labelled all data as scrolling. This process eliminated the need for manual labelling. The data transferred to the computer included the timestamp, nine sets of sensor data (x, y, z-axis for accelerometer, gyroscope and magnetometer) and the labelled smartphone activity.

The raw data from smartphone sensors is time-series data and it would not be feasible to put such a large amount of data into the machine learning model at once. Therefore, a sliding-window is used to segment the data into sections, where each section has the same amount of data. Each of the segmented windows was used to classify an instance of the smartphone activity. Based on previous ranges of suitable window sizes for similar investigations [8,9,10,12,14,16], we systematically explored sizes of 2.56 s, 5 s, and 8 s. Consecutive windows had an overlap of 50%. Sampling frequencies can have an impact on the performance of the classification. Further, power consumption differs across sampling frequencies. To examine the effect of different sampling frequencies on the model’s performance, we tested the two sampling frequencies which the smartphone can provide, which are 5 Hz and 50 Hz. In our study, 9 participants’ data was sampled with a frequency of 5 Hz, and the remaining 12 participants were sampled with a frequency of 50 Hz. The sensor data from these participants for reading, scrolling, typing, watching and idle, while sitting and walking, is included in the Appendix A.

### 3.2. Feature Extraction And Classification

#### 3.2.1. Feature Extraction

Feature extraction plays an essential role in the performance of the smartphone activity recognition system. This process allows useful information to be extracted from the raw data which can then be used to improve the performance and computation time of the machine learning algorithm.

Mourcou et al. compared Apple and Samsung Galaxy smartphone IMU sensors with gold standard industrial robotic arm IMU sensors and found them to be comparable [34]. This research supported the use of the raw data directly for feature extraction without any pre-processing filters.

Time and frequency domain features were selected. We extracted the following features from our raw IMU sensor data: mean, standard deviation, variance, mean absolute deviation, minimum, maximum, inter-quartile range, average resultant acceleration, skewness, kurtosis, signal magnitude area, energy, zero-crossing rate, and the number of peaks of the data in the window. The description of the features is shown in Table 2. The number of parameters indicates the number of parameters related to each feature. There are nine sets of data related to means (one for each axis of each sensor) and three sets of data related to average resultant acceleration (one for each type of sensor).

A correlation table for the features was generated using Pearson’s Correlation Coefficient. The correlation coefficient helps to identify the strength of association between two features. Any two features (i.e., a pair) with a high correlation coefficient absolute value are highly related, meaning one cannot provide extra information to the algorithm if the other feature is already present. The presence of both highly correlated features would affect the overall weighting of features, increasing bias and therefore reducing algorithm accuracy. We examined feature pairs for absolute correlations of greater than 0.9 for removal, as these are classified as strongly related pairs [35]; one pair was removed from the feature list. The comparison of the correlation table before and after removing these features is shown in Figure 5. There are 71 features in the final list, which includes standard deviation, variance, skewness, kurtosis, zero-crossing rate and the number of peaks for all three types of sensors, mean for the accelerometer and gyroscope and minimum, maximum, average resultant acceleration, signal magnitude area and energy for the accelerometer.

These features are further analyzed by completing the temporal autocorrelation, which is the relationship between the successive values of the same feature. As shown in Figure 6, the autocorrelation of the features can be categorized into three types: linear, periodic and no correlation. The linear correlation means that the values of a feature are related to each other, so the closer two values are in the timeline, the greater the correlation between them. Periodic correlation occurs when the values of a feature are repeated in some frequency. Therefore, the value of an instance would be highly correlated to values from some time units before. Lastly, no correlation means that the previous value of a feature does not impact on the current value. This result shows how the sampling frequency can have a significant impact on the performance of the algorithm, as critical information could be lost if the sampling frequency is not high enough. To investigate the impact of sampling, nine participants’ data were collected with a sampling frequency of 5 Hz, and the rest were collected with 50 Hz.

#### 3.2.2. Activity Classification

Selected features are put into machine learning models as training sets and are used for real-time classification. We used the “scikit learn” python package for training the classifiers. We trained and evaluated seven different classification algorithms: Multi-Layer Perceptron (MLP), Support Vector Machine (SVM), K-Nearest Neighbor (KNN), Bootstrap Aggregating (Bagging), Adaptive Boosting (AdaBoosting), Random Forest (RF) and Extremely Randomized Trees (also called ExtraTree, ET).

We used the k-fold cross-validation technique to evaluate the performance of the algorithms. To minimize the bias between participants, no participant’s data appears in two folds. Therefore, 5-fold cross-validation was used because the data we collected from the participants was not enough to perform the 10-fold cross-validation (nine participants’ data was collected in 5 Hz). The data used for training the model is divided into five sets, where each set consists of data from two participants on average. Among the five sets, one of them is selected to be the test set, and the remaining four sets are used to train the classifier; the accuracy is then evaluated on the test set. This process is iterated for each of the sets, and the final validation result is the average of all iterations. We aimed to classify the human smartphone activities into 10 labels—sitting scrolling, sitting reading, sitting typing, sitting watching, sitting idle, walking scrolling, walking reading, walking typing, walking watching, and walking idle.

We examine three population algorithms for classifying human activities—the Multi-Layer Perceptron (MLP), Random Forest (RF) and Extremely Randomized Trees (ET).

Multi-Layer Perception is a feedforward artificial neural network (ANN) for supervised classification [36]. This algorithm consists of interconnected nodes which can be divided into at least three layers (an input layer, one or more hidden layers, and an output layer). Every connection between nodes has a weight, and every node in the system except input nodes is associated with an activation function which determines whether a signal should be fired or not based on the incoming signal and its weight. The output of the algorithm is determined by how the signal propagates through the nodes. The model is trained using the backpropagation technique. The advantage of using Multi-Layer Perceptron is that it has the ability to learn complex, non-linear relationships, and it is generalizable; and there is not restriction on how the data is distributed [36].

The Random Forest and Extremely Randomized Trees algorithms are similar. Random Forest is an ensemble algorithm which constructs a large number of decision trees where each tree node splits on a random subset of features [37]. The Extremely Randomized Trees algorithm is also a tree-based ensemble method for supervised classification and solving regression problems [38]. There are two major differences between these two algorithms. The first is that Extremely Randomized Trees algorithm does not apply bagging for its training data; it uses the original dataset for all decision trees. In contrast, Random Forest samples the original dataset with replacement to generate new datasets for each of the trees. The second difference is the tree node splitting method. For the Random Forest algorithm, the split of the tree node is based on the best feature among a random subset of all features of the data, and it uses a majority vote to determine the final classification result. In contrast, the Extremely Randomized Trees algorithm splits each tree node with a random number of features and randomized cut-points [38].

The advantage of using the Random Forest algorithm is that it is an ensemble model. These usually produce higher accuracy than individual algorithms. They also introduce randomness into the algorithm to reduce the bias, which further improves its accuracy.

Due to the bootstrap sampling method and the randomness introduced in tree nodes splitting, the corresponding nodes in different trees usually cannot split on the same feature. These two criteria ensure the diversity of the individual trees, which significantly reduces error rate, allowing the majority vote of trees to provide a more accurate result. The Extremely Randomized Trees algorithm maintains most of the properties of the Random Forest algorithm but it minimizes its variance by increasing the bias [38]. The Extremely Randomized Trees algorithm is faster than the Random Forest, and as the size of the dataset increases, this difference increases. This efficiency property is ideal for mobile devices as the computational power required to perform classification is limited.

## 4. Results

In total, 207 blocks of 2-minute activity data were recorded from the 21 participants. Three blocks were invalid due to hardware failure. Several different window sizes and sampling frequencies were tested using 5-fold cross-validation. The results are shown in Table 3. The performance of the MLP, RF and ET algorithms were very close to each other, and they always outperformed the other algorithms, and the results from the two sampling frequencies were very close to each other. Therefore, we chose to consider these three algorithms for the rest of the study. From the Table 3, it is observed that the longer the window size, the better the performance. To test whether this is true for even longer window sizes, we did further tests on window sizes of 10 s, 15 s and 20 s, as shown in Table 4. We noticed that for 50 Hz data, optimal accuracy is likely to occur when the window size is around 15 s. For the three good algorithms we found to be the best (MLP, RF and ET), the accuracy decreases at 20 s (best accuracy of 67.7%±2.8% occurs when using MLP with a 15-s window size). And for the 5 Hz data, the optimal window size occurs beyond 20-s, which is too long to be used for real life application. We note that recent research on media multitasking demonstrates that users engage in continuous digital tasks for on average 11 s before switching attention [39]. Thus users will perform the activities we classified (i.e., reading, watching) for more than five seconds, which means a five second recognition delay would not be problematic. Therefore, we propose that five seconds is an ideal window size to balance accuracy with responsiveness in our study. With the five-second window size, 4639 samples of data were extracted from the 50 Hz data, and 3620 samples from the 5 Hz data.

The normalized confusion matrix was evaluated for each of the three models using data from two frequencies with a window size of 5 s; this is shown in Figure 7. The confusion matrix is a table which summarizes the classification result for each label; it breaks down the correct and incorrect predictions for each label related to other labels to help visualize how the model makes predictions. Figure 7 shows that all models have difficulties in distinguishing between reading and scrolling activities; this occurs in both the sitting and the walking condition. This difficulty is likely due to the similarities in the gestures involved in the activities. Therefore, to improve the accuracy of our results, we decide to combine these two labels into one label.

### 4.1. Reducing Labels

Due to the similarity in gesture motion between reading and scrolling, we decided to combine these two activities into one, and only use label “scrolling” for both activities. Both of these activities use a vertical swipe gesture on the smartphone, but usually with different frequencies; from observation, the frequency of vertical swipe while reading is generally lower than scrolling. After the label of reading was replaced by scrolling, the performance of the models was improved, as shown in Table 5. The table shows that in general, a higher sampling frequency can provide a better result. This was further confirmed by the confusion matrix of the models after we combined the labels, as shown in Figure 8. With a lower sampling frequency, the model would have difficulty in classifying activities between scrolling and typing, especially when the user is walking. Therefore, to achieve higher accuracy, a higher sampling rate is necessary, which means it is not feasible to maintain a reasonable accuracy and reduce power consumption by using lower sampling frequency. To determine which of the three models has the best performance, we compared precision, recall and F1-scores (Table 6). The Extremely Randomized Trees algorithm has the highest precision, recall and f1-score.

### 4.2. Reading Quality

Short multiple choice questions were added after each reading task to motivate the participant to pay attention to reading. Each participant’s answers to the multi-choice questions were recorded and analyzed to help determine the quality of reading. The results of the questions are shown in Table 7. This table shows how the participants performed under different conditions. For example, nine participants read article 1 under the sitting condition, and five of them answered the question correctly. The table was based on the question answers from 18 participants. The data from three participants was not valid because of technical issues. Due to the randomization of condition order and article orders, the number of participants under each condition with each article in the reading task was different. The results show a higher reading quality for the sitting condition than in the walking condition. This is expected as under the sitting condition there is less distraction and less for the participant to focus on. During the walking condition, the participants needed to pay attention to the environment while reading the articles. The study facilitators noticed a significant reduction in speed when participants performed activities while walking. Participants tended to walk more slowly than what would be considered typical while typing or reading articles.

### 4.3. Self-Reported Frequency of Engaging in Smartphone Activities

The questionnaire response data found differences in the frequency with which participants engaged in scrolling, reading, typing and watching videos, with a lower frequency of video watching compared to other activities (Table 8). Most participants reported that they performed the activities at least daily (i.e., scrolling feeds, reading, texting, and watching videos on a smartphone). More than 90% of the participants read, texted and scrolled feeds at least daily, whereas only 52% watched videos on a smartphone at least daily. Approximately half of the participants reported engaging in reading, texting and scrolling while walking on a daily basis, and only one participant watched videos on their smartphone while walking on a daily basis.

## 5. Discussion

We have conducted a study with human participants to investigate the accuracy with which IMU sensor data can be used to classify the typical activities that people perform on smartphones: scrolling, typing, and watching videos while seated or walking. Using the Extremely Randomized Trees algorithm, we achieved an activity recognition accuracy of 78.6%. To further validate the performance of the model, we performed the 5-fold cross-validation on the 5 Hz data with the same setting, which resulted in an accuracy of 75.0%. In order to use all the data we collected from 21 participants, we subsampled the 50 Hz data, and produced a higher cross-validation score of 78.2%, with a lower standard deviation. An increase in dataset size reduces the impact of outliers and unusual data, thus reducing the bias of the dataset and resulting in an increase in accuracy. Thus, it is reasonable to conclude that the performance of our model would increase with a larger dataset.

To explore whether the data we collected was sufficient to train the model, an accuracy vs sample size plot was created for each of the three models, as shown in Figure 9. The plots suggests that the accuracy of the model became fairly stable when the sample size was greater than 1500. This supports that the amount of data we collected was sufficient to produce a valid result.

The accuracy of our model is only fair and we note that it is lower than some prior studies. The movement involved in the general smartphone activities which we targeted is small, as phones do not move excessively when people are scrolling, typing or watching videos. These activities are physically smaller than the activities which were classified in the prior studies. They include general physical activities which involve large movements of the entire body such as walking [8,9,10,11,12,13,14,15,16,17], ascending and descending stairs [8,9,10,11,13,14,15,16,17], jogging (fast walking) [11,13,16,17], running [10,11,12,15,17], biking [12,16], standing [8,9,10,12,13,14,16,17], sitting [8,9,10,12,13,14,15,16,17], and lying [8,9,14], in addition, Hassan et al. also includes transition between the activities (for example moving from sitting to standing) [8], Bayat et al. includes aerobic dancing [11], Shoaib et al. includes other activities such as eating, typing, writing, smoking, drinking coffee and giving a talk [16], and lastly, Lu et al. includes several race walking and basketball playing activities [17]. Some prior work used multiple devices and additional sensors for data collection [15,16,17] and restricted the placement of the devices [9,11,12,13,14,15,17] (for example, placing smartphone in trousers pocket) to reduce variations in the data in order to improve the performance. In contrast, we did not give specific instructions to participants as we wanted them to perform activities in typical ways. Our research is lightweight, and only uses IMU sensors for recognition, and we did not restrict how the participants used the smartphone. Our work demonstrates that we can achieve reasonable performance in tracking smartphone activities on an “unrooted” smartphone using only IMU sensors.

In our study, we found a similarity between the reading and scrolling activities. We argue that it is reasonable to classify both activities as scrolling because they are similar except for the frequency of the vertical swipe gesture and the range of movement. This similarity is reflected in the confusion matrix, where there is a high probability of a reading activity being classified as a scrolling activity (and vice versa).

Our model includes typical activities done on smartphones. We initially considered other activities such as playing a mobile game, reading an eBook and side-swiping (for e.g., a digital carousel or a photo album) but these were excluded. We did not select gaming as one of our activities because smartphone games can have enormous variation in the types of movements and gestures that are performed. Different games can have a different controlling method; for example, Flappy Bird only requires tapping. In contrast, driving simulation games can require placing the phone in landscape, and rotating the phone during the game to control the car. Similarly to playing games, there are three major types of gestures which can be used while reading an eBook: vertical swipe, side-swipe or tapping, and different users or applications may have different preferences for which is used. Side-swipe gestures, in general, were found to be less common, especially when the user needs to perform side swipe for more than 5 s. As a result, our final classification was based on four typical activities: scrolling, typing, watching and non-use. Each of these activities was performed under the two conditions, sitting and walking.

We had to discard some data due to technical difficulties and in situations when participants acted outside the scope of our study, invalidating their data. Only a small proportion of data was discarded due to technical issues. Once the test phone was accidentally disconnected from the laptop, which prompted the necessity to restart the Android application. Since the order of the contents was randomly generated, the participant might be presented with the same content twice, this may have affected their behavior and data.

At the current stage, the processing power of most smartphones is capable of performing this kind of recognition on its own by deploying the machine learning model to the smartphone. However, the high computational power required by the process would drain the battery, which may not be feasible for running the recognition for a long time. One possible solution would be to send the heavy computational tasks to a remote server to reduce the battery drain on the smartphone and would extend the endurance of the phone. One other possible solution would be to only turning on the recognition on during critical periods.

### Limitations

Limitations of this research include the small sample of participants, the restricted number of labels used in the classification, and the physical environment involved in the study. We only recruited 21 participants. Given that we separated the data collection into two different frequencies, we would need more data to maximize accuracy. Since all of the participants recruited were university students, and most studied computer science or software engineering, the result of classification may not represent the more general population. More participants need to be recruited from more varied backgrounds and fields to produce a more generalizable result. As described above, the labels classified do not include gaming or game-related activities. However, this limitation is unlikely to be overcome as the gestures used in gaming can overlap with the gestures used in other activities. One of the possible ways to detect whether a person is gaming is to use facial recognition or sensors which collect physiological signals of the human body in order to classify the person’s emotion. The system can be combined with smartphone IMU sensors to predict the user’s smartphone activity. Activities performed in the walking conditions were performed as participants walked around a foyer. Participants had to walk in a circuit more than ten times, which may not be representative of a real-world environment.

## 6. Conclusions and Future Directions

In this paper, we reported on a study which shows how accurately smartphone IMU sensors can classify typical smartphone activities. We developed an Android application which collected IMU sensor data and used it to train a machine learning model for smartphone activity classification. The classifier can distinguish four different activities (scrolling, typing, watching videos, non-use) under two different conditions (sitting and walking) with an accuracy of 78.6% using the Extremely Randomized Trees algorithm.

To improve this research and make the results more realistic, more participants need to be recruited from different user groups (i.e., different occupations and different age groups). Our future research will include the collection of sensor data with participants using their smartphones naturalistically rather than in a prescribed experiment. To do so, we would develop an app which collects sensor data and that sends it to a remote server in the background while the user performs their usual activities. We can explore how to involve the user in improving the labeling and reliability of the data. For instance, the smartphone app could display how users’ behavior was labeled, and enable them to correct those labels if necessary.

Another important direction for further research is to investigate the performance of the models for long-term single person use. Due to the behavioral differences between users, the model is not specific for a single person. The accuracy of the model can be improved by training on a sufficient amount of data from a single person, which would allow the classification to be more personalized and more accurate for that person.

Our research provides insight into the question of whether typical smartphone activities can be recognized using standard internal smartphone sensors. Such recognition could provide benefits such as personalization based on smartphone activities, but also indicates risk for user monitoring via IMU sensors.

## Figures and Tables

**Figure 1 sensors-20-00655-f001:**
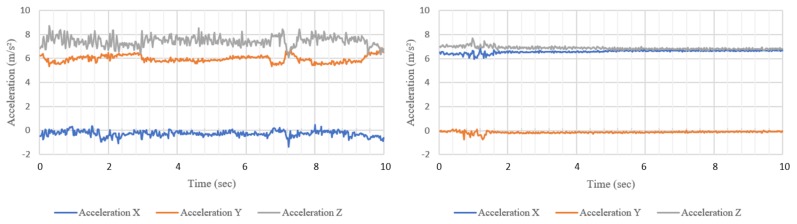
Ten seconds of acceleration data for a. typing (left) and b. watching (right), both while seated. The acceleration data for typing shows greater variation while the data for watching is smoother.

**Figure 2 sensors-20-00655-f002:**
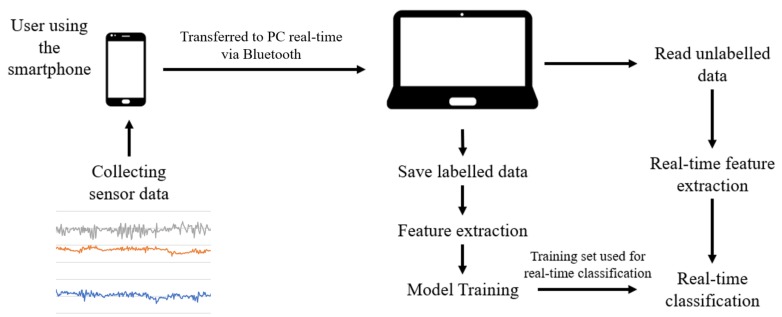
Study data collection and processing workflow: a user uses the smartphone for specified time periods and specified activities, while sensor data is recorded. In real time, data is transmitted and classified.

**Figure 3 sensors-20-00655-f003:**
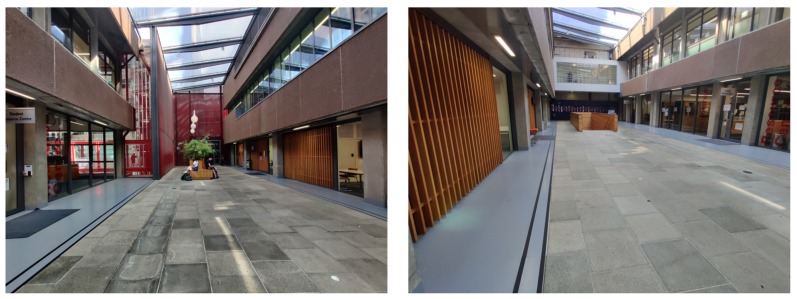
Foyer of the building used for the walking condition, selected to simulate the real walking conditions that smartphone users might experience in daily life.

**Figure 4 sensors-20-00655-f004:**
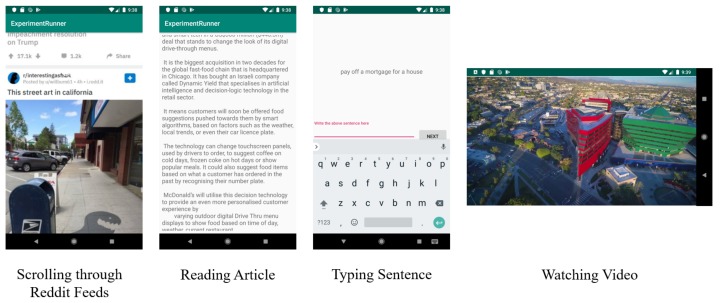
Screenshots of the four tasks which participants completed as part of the study, while software collects the IMU sensor data. Each of the four tasks is completed while sitting and while walking.

**Figure 5 sensors-20-00655-f005:**
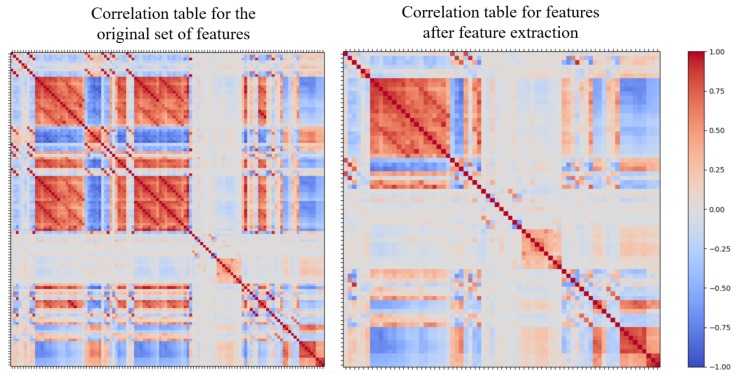
Correlation table before and after feature extraction.

**Figure 6 sensors-20-00655-f006:**
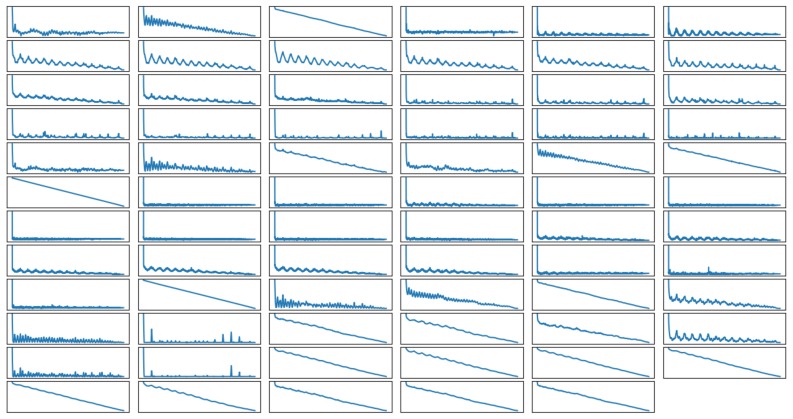
Autocorrelation for the final selected features.

**Figure 7 sensors-20-00655-f007:**
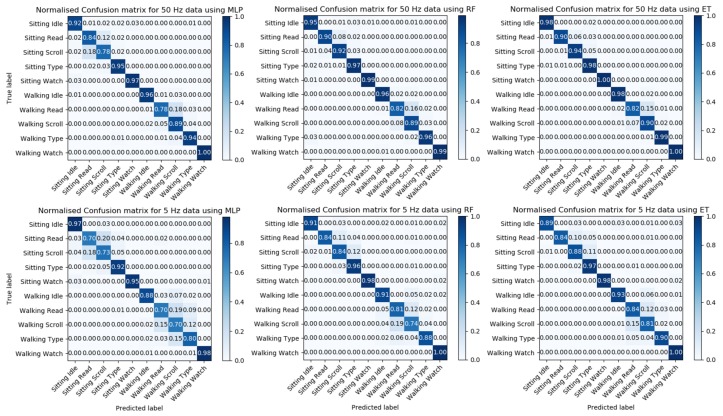
Confusion matrix for the three models (Multi-Layer Perceptron (MLP), Random Forest (RF) and Extremely Randomized Trees (ET)) with different frequencies.

**Figure 8 sensors-20-00655-f008:**
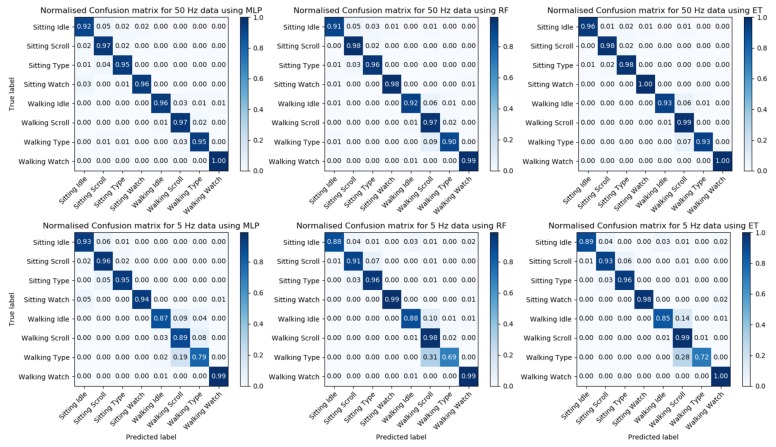
Confusion matrix for the three models (MLP, RF and ET) with different frequencies, after combining labels.

**Figure 9 sensors-20-00655-f009:**
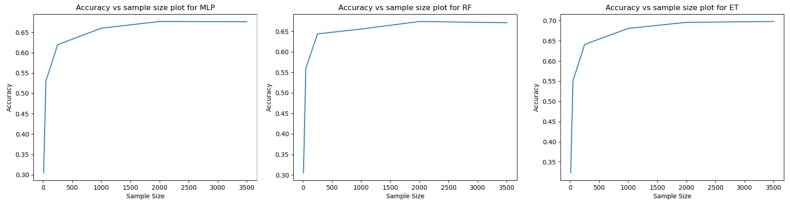
Accuracy vs sample size plot for the three models (MLP, RF and ET).

**Table 1 sensors-20-00655-t001:** Sensors and features used in prior research on human activity recognition with smartphones.

Sensors	Features	Reference
Accelerometer, Gyroscope	Mean, Standard Deviation, Mean Absolute Deviation, Window Maximum Value, Window Minimum Value, Frequency Skewness, Maximum Frequency, Average Energy, Signal Magnitude Area, Entropy, Window Interquartile Range, Pearson Correlation Coefficients, Frequency Signal Weighted Average, Spectral Energy and Angle between a Central Vector and Mean of Three Consecutive Windows	[8]
Accelerometer, Gyroscope		[9]
Accelerometer, Gyroscope, Magnetometer	Mean, Average Absolute Difference, Standard Deviation, Average Resultant Acceleration and Histogram	[10]
Accelerometer	Mean, Elapse Time between Consecutive Local Peaks, Average of Peak Frequency (APF), Variance of APF, Root Mean Square, Standerd Deviation, Minmax Value and Correlation	[11]
Accelerometer, Gyroscope, Magnetometer	Mean, Variance, Standard Deviation, FFT Coefficient, Zero Crossing Rate, Maximum Correlation Value and Index of Max Correlation	[12]
Accelerometer	Mean, Standard Deviation, Average Absolute Difference, Average Resultant Acceleration, Time between Peaks and Binned Distribution	[13]
Accelerometer, Gyroscope	Mean, Standard Deviation, Median Absolute Value, Window Maximum Value, Window Minimum Value, Signal Magnitude Area, Energy, Interquartile Range, Entropy, Autoregression Coefficient, Correlation, Maximum Frequency Index, Mean Frequency, Skewness, Kurtosis, Energy Band, and the Angle between Two Vectors	[14]
Accelerometer	Autoregressive Coefficients and Signal Magnitude Area	[15]
Accelerometer, Gyroscope, Magnetometer	Mean, Standard Deviation, Magnitude, Window Maximum Value, Window Minimum Value, Semi-quartile, Median and Sum of the First Ten FFT Coefficient	[16]
Accelerometer	Mean, Standard Deviation, Variance, Skewness, Kurtosis, Correlation and Signal Magnitude Area	[17]

**Table 2 sensors-20-00655-t002:** List of features and description for the investigation of recognition of smartphone activities with inertial measurement unit (IMU) sensors.

Features	Number of Parameters	Description
Mean	9	The average value of the data for each axis in the window
Standard Deviation	9	Standard deviation of each axis in the window
Variance	9	The square of the standard deviation of each axis in the window
Mean Absolute Deviation	9	The average difference between the mean and each of the values for each axis in the window
Window Minimum Value	9	The minimum value of the data for each axis in the window
Window Maximum Value	9	The maximum value of the data for each axis in the window
Inter-quartile Range	9	The range of the middle 50% of the values for each axis in the data
Average Resultant Acceleration	3	The average of the square roots of the sum of the squared value of 3 axis for each type of sensor in the data
Skewness	9	The degree of distortion of each axis from the symmetrical bell curve in the window
Kurtosis	9	The weight of the distribution tails for each axis in the window
Signal Magnitude Area	3	The normalized integral of 3-axis for each type of sensor in the window
Energy	9	The area under the squared magnitude of each axis in the window
Zero Crossing Rate	9	The number of times the data crossed the 0 value for each axis in the window
Number of Peaks	9	The number of peaks for each axis in the window

**Table 3 sensors-20-00655-t003:** Cross-validation accuracy and standard deviation for the classification algorithms trained for different window sizes.

Algorithms	50 Hz with 2.56 s Window Size	50 Hz with 5 s Window Size	50 Hz with 8 s Window Size	5 Hz with 2.56 s Window Size	5 Hz with 5 s Window Size	5 Hz with 8 s Window Size
MLP	60.3%±2.9%	62.2%±3.1%	64.5%±3.1%	60.3%±3.8%	62.8%±3.5%	64.6%±4.3%
SVM	36.9%±5.3%	36.3%±4.3%	35.1%±4.4%	34.4%±12.2%	33.0%±13.7%	31.8%±12.2%
KNN	44.5%±4.5%	43.4%±5.2%	42.7%±4.6%	40.0%±7.7%	38.5%±7.3%	37.8%±5.5%
Bagging	56.0%±7.1%	55.9%±4.7%	58.5%±6.4%	57.4%±3.7%	57.9%±6.8%	56.7%±7.6%
AdaBoosting	37.3%±2.1%	38.1%±7.8%	27.0%±6.1%	33.9%±3.9%	31.4%±10.0%	31.2%±10.5%
RF	58.2%±7.2%	60.0%±8.5%	61.0%±8.8%	61.3%±6.2%	61.8%±5.6%	63.1%±6.2%
ET	60.4%±5.4%	61.8%±6.2%	62.6%±5.4%	62.6%±5.7%	63.8%±6.0%	64.6%±5.7%

**Table 4 sensors-20-00655-t004:** Cross-validation accuracy and standard deviation for the classification algorithms trained for longer window sizes.

Algorithms	50 Hz with 10 s Window Size	50 Hz with 15 s Window size	50 Hz with 20 s Window Size	5 Hz with 10 s Window Size	5 Hz with 15 s Window Size	5 Hz with 20 s Window Size
MLP	65.5%±3.8%	67.7%±2.8%	67.0%±4.5%	64.8%±5.3%	64.8%±5.5%	66.3%±3.6%
SVM	34.9%±4.6%	32.9%±5.6%	33.5%±5.0%	31.8%±11.8%	33.5%±11.9%	33.2%±14.2%
KNN	41.1%±5.3%	39.4%±4.0%	35.7%±5.0%	37.3%±5.7%	37.7%±5.6%	37.9%±3.9%
Bagging	55.9%±4.3%	54.9%±3.7%	57.7%±4.6%	56.4%±4.1%	57.7%±6.3%	56.5%±5.2%
AdaBoosting	28.6%±7.8%	28.9%±12.2%	34.9%±10.1%	21.1%±2.5%	23.0%±6.5%	22.7%±4.5%
RF	61.1%±7.6%	60.3%±8.6%	61.0%±9.2%	63.5%±5.8%	65.6%±6.8%	65.4%±7.7%
ET	62.6%±5.2%	63.8%±6.0%	63.0%±5.5%	65.6%±6.2%	65.8%±6.5%	66.6%±5.2%

**Table 5 sensors-20-00655-t005:** Cross-validation score for the three models (MLP, RF and ET) with different frequencies.

Algorithms	All Labels	Combined Reading and Scrolling
MLP with 50 Hz data	62.2%±3.1%	75.4%±2.6%
RF with 50 Hz data	60.0%±8.5%	77.0%±6.5%
ET with 50 Hz data	61.8%±6.2%	78.6%±5.3%
MLP with 5 Hz data	62.8%±3.5%	75.0%±4.5%
RF with 5 Hz data	61.8%±5.6%	73.7%±5.5%
ET with 5 Hz data	63.8%±6.0%	74.7%±4.6%

**Table 6 sensors-20-00655-t006:** Precision, recall and F1-score for MLP, RF and ET.

Algorithms	Precision	Recall	F1-score
MLP with 50 Hz data	0.961	0.959	0.960
RF with 50 Hz data	0.963	0.952	0.957
ET with 50 Hz data	0.978	0.971	0.974
MLP with 5 Hz data	0.922	0.915	0.918
RF with 5 Hz data	0.931	0.908	0.915
ET with 5 Hz data	0.940	0.913	0.921

**Table 7 sensors-20-00655-t007:** Results of the questions after reading task as an approximation of reading quality.

	Article 1	Article 2	Article 3	Correctness
Sitting Condition	Correct: 5, Incorrect: 4	Correct: 2, Incorrect: 1	Correct: 4, Incorrect: 2	61.1%
Walking Condition	Correct: 1, Incorrect: 3	Correct: 2, Incorrect: 6	Correct: 2, Incorrect: 4	27.8%
Correctness	40.3%	50%	50%	44.5%

**Table 8 sensors-20-00655-t008:** Frequencies of participants’ smartphone activities.

Activities	Constantly throughout the Day	A Few Times a Day	Daily	A Few Times per Week	Once a Week or Less
Typing	38.1%	42.9%	14.3%	4.8%	0%
Reading	42.9%	38.1%	14.3%	4.8%	0%
Watching video	4.8%	33.3%	14.3%	28.6%	19.0%
Scrolling news feeds	33.3%	33.3%	23.8%	4.8%	4.8%
Typing while walking	4.8%	38.1%	19.0%	23.8%	14.3%
Reading while walking	14.3%	14.3%	23.8%	28.6%	19.0%
Watching video while walking	0%	4.8%	0%	28.6%	66.7%
Scrolling news feeds while walking	9.5%	4.8%	28.6%	28.6%	28.6%

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
