# Peer review of "Real-time Smartphone Activity Classification Using Inertial Sensors—Recognition of Scrolling, Typing, and Watching Videos While Sitting or Walking"

_sensors, 2020, doi:10.3390/s20030655_

Round 1
Reviewer 1 Report
This paper presented a comparison of methods for detecting cell phone gestures using sk-learn, achieving an accuracy of 78.6 %.
Overall the paper is ok, however the usefulness of the outcomes is not persuasive. For example, one could directly detect things like mood from the acceleration dtaa and would not first need to detect swiping or other phone gestures. Please make a better argument for real use acases of this classification.
The term activity recognition is misleading, because it normally refers to activities such as walking or sitting. Please qualify it, especially in the title and abstract and conclusion. e.g. smart phone gesture recognition.
Furthermore, there is a lack of literature search into smart phone gesture recognition - there are many papers related to security of smart phones where they show they can detect passwords with accleration data:
R. Ning, C. Wang, C. Xin, J. Li and H. Wu, "DeepMag: Sniffing Mobile Apps in Magnetic Field through Deep Convolutional Neural Networks," 2018 IEEE International Conference on Pervasive Computing and Communications (PerCom), Athens, 2018, pp. 1-10.
doi: 10.1109/PERCOM.2018.8444573
5 seconds is a very long window for the recognition lag when detecting gestures, for the use cases specified, so smaller windows need to be investigated or a use case presented where a 5 second detection lag would not be problematic.
The number of participants and situations are very limited, in the outlook please specify more clearly how one could expand the study for more realistic and larger data colelctions. And how such a dataset could be labelled.
Finally, as there are so many results, please distill them and make clearer the most important ones and move the rest to a supplement. Summary values such as the F1-scores and the miss and detection rates would be useful here.
Author Response
We thank the reviewer for saying that our paper was “overall” "ok." We summarize each of the reviewers comments as a subheading, and respond to each one in turn. We have made commensurate major changes to the manuscript, which strengthened it.
Usefulness of outcome.
We outline two new arguments for this investigation in our introduction, and include an additional four citations for framing. We thank the reviewer for the reference on security as this is an important line of argumentation, along with personalization. We add the following to the introduction:
"Human activity tracking using smartphone sensors has been a key use case since smartphones have been released, in part due to smartphones' physical size, ubiquity, unobtrusiveness and ease of use. Whole-body human activities such as walking and running have been investigated but physically smaller human activities such as reading or typing have not been explored. Knowing whether a user is scrolling, typing or watching videos is valuable to better understand users. Digital content and experience providers benefit by tracking users' smartphone activities to inform personalization, as demonstrated by several recent patents ~\cite{hauser2011digital, linden2011content, smith2005content}. For example, news and social media applications log when users watch videos or read in order to present personalized content that is more likely to be engaging to each user ~\cite{thurman2011making}. If a news or social media app uses inertial measurement unit (IMU) sensors to learn that a user watches videos or reads when using other apps on their smartphone, they can present the user with more personalized content (i.e., more videos or more text, depending on what the user uses and likes).
The other side of this investigation is a question of security. It is not currently possible to access user action logs directly, due to smartphone data access privileges. Detailed logs of all user smartphone activities may be considered private or sensitive. As of 2019, smartphones do not ask users for permission to access IMU sensor data. A phone would need to be "rooted", in other words, an application would need to gain root access control, in order to access activity data directly. This may be because current IMU applications do not seem as privacy-sensitive as other permission-protected sensors such as Global Positioning System (GPS), the microphone and camera. However there is growing evidence that IMU sensors can be used for nefarious purposes, such as location tracking \cite{narain2016inferring} and password sniffing~\cite{ning2018deepmag}."
The term “activity recognition”.
We agree that "human activity recognition on smartphones" tends to refer to activities such as walking and thus we change to "human smartphone activity recognition on smartphones" throughout the paper. We appreciate the suggestion for “gesture recognition” however that may also be confusing because we are looking at human activities of reading rather than lower-level gestures such as pure swiping.
Window size.
With the above point, we further clarified the human smartphone activities that we are interested in, which are reading, watching. As these humans typically engage in these smartphone activities for longer than 5 seconds, we hope it now makes sense that we selected 5 second window as appropriate for these. Further, we added citations of prior work that used similar window sizes:
“Based on previous ranges of suitable window sizes for similar investigations \cite{hassan2018robust, ronao2016human, voicu2019human, ustev2013user, anguita2013public, shoaib2016complex}, we systematically explored sizes of 2.56 seconds, 5 seconds, to 8 seconds.”
Sample size.
A sample of 21 participants is not unusual for this type of research, though we agree that this is a limitation.
Future work.
We expand upon future work:
“To improve this research and make the results more realistic, more participants need to be recruited from different user groups (i.e., different occupations and different age groups). Our future research will include the collection of sensor data with participants using their smartphones naturalistically rather than in a prescribed experiment. To do so, we would develop an app that collects sensor data and that sends it to a remote server in the background while the user performs their usual activities. We can explore how to involve the user in improving the labeling and reliability of the data. For instance, the smartphone app could display how users’ behaviour was labeled, and enable them to correct those labels if necessary.”
Clarifying the most important results.
To highlight our most important results, we removed the analysis and discussion of the cognitive load condition, as it was less central to our work and did not perform well. This shortened the manuscript from 20 to 18 pages.
To highlight our final result, we added a table with f1 scores and the following text:
“To determine which of the three models has the best performance, we compared precision, recall and F1-scores (Table \ref{tab:f1}). The extremely randomized trees algorithm shows the highest precision, recall and f1-score.”
We thank the reviewer for their consideration. We clarified the usefulness of the investigation, streamlined our manuscript by removing unnecessary text, and improved the results with f1 scores. We have done a thorough editing pass of the paper the fix language issues. We appreciate the opportunity to have made these extensive changes which have significantly improved the manuscript.
Reviewer 2 Report
This paper proposes an activity classification method using smartphone sensors. Some comments are listed in the following.
The major concern is whether the task done in this paper is meaningful or useful. In the beginning of Section Introduction, “it is not currently possible to do so directly due to smartphone data access privileges. A phone would need to be “rooted”, in other words, an application would need to gain root access control, in order to access activity data directly”. The task can be easily solved by the phone producer to provide such information, which is the ‘groundtruth’ not a predicted value (not sitting or walking, but scrolling and typing). In other words, if this task is really useful, the phone will provide true data. If not useful, this paper is meaningless. On the other hand, there are activity recognition based on the phone sensors, but can not be provided by the phone, for example, walking, sitting, running as in the SHL challenge. Such classification is always useful regardless of the phone information.
http://www.shl-dataset.org/activity-recognition-challenge-2019/
As in section 4 result, the tsfresh method is not used. Then what is the purpose of describing it? The authors should test its performance in comparison with the manual setting to show whether the manually selected feature include an enough number of features to do the classification.
Fig. 5 is not very clear. In the second correlation table, there are also highly correlated features existing. Is the correlation in the table normalized?
Typo: Line 360 includes ??.
Overall, from the perspective of new methods, the paper shows no novelty. From the perspective of new task, the authors are suggested to highlight its importance.
Author Response
We thank the reviewer for their comments. We summarize each of the reviewer's comments as a subheading, and respond to each one in turn. We have made major changes to the manuscript, which further strengthened it.
Importance of task.
We agree that the smartphone can provide the ground truth of this data, however that data is currently protected. We clarify why the investigation of smartphone activity recognition via IMU sensors is useful in the introduction:
"Human activity tracking using smartphone sensors has been a key use case since smartphones have been released, in part due to smartphones' physical size, ubiquity, unobtrusiveness and ease of use. Whole-body human activities such as walking and running have been investigated but physically smaller human activities such as reading or typing have not been explored. Knowing whether a user is scrolling, typing or watching videos is valuable to better understand users. Digital content and experience providers benefit by tracking users' smartphone activities to inform personalization, as demonstrated by several recent patents ~\cite{hauser2011digital, linden2011content, smith2005content}. For example, news and social media applications log when users watch videos or read in order to present personalized content that is more likely to be engaging to each user \cite{thurman2011making}. If a news or social media app uses inertial measurement unit (IMU) sensors to learn that a user watches videos or reads when using other apps on their smartphone, they can present the user with more personalized content (i.e., more videos or more text, depending on what the user uses and likes).
The other side of this investigation is a question of security. It is not currently possible to access user action logs directly, due to smartphone data access privileges. Detailed logs of all user smartphone activities may be considered private or sensitive. As of 2019, smartphones do not ask users for permission to access IMU sensor data. A phone would need to be "rooted", in other words, an application would need to gain root access control, in order to access activity data directly. This may be because current IMU applications do not seem as privacy-sensitive as other permission-protected sensors such as Global Positioning System (GPS), the microphone and camera. However there is growing evidence that IMU sensors can be used for nefarious purposes, such as location tracking \cite{narain2016inferring} and password sniffing~\cite{ning2018deepmag}."
To focus on this point of importance, and to streamline the document, we remove the discussion of cognitive load, as this aspect was not central to the contribution.
Novelty.
"Human activity recognition on smartphones" tends to refer to activities such as walking which have been investigated. We investigate "human smartphone activity recognition on smartphones", and have clarified this difference throughout the paper.
Tsfresh.
We agree that the tsfresh is not needed. We removed it.
Figure 5.
We have used a standard visualisation and note that the correlations are less than 0.9, which is a standard threshold. We include a new reference for support, and we clarify this with the following text:
“We examined feature pairs for absolute correlations of greater than 0.9 for removal, as these are classified as strongly related pairs \cite{akoglu2018user}, and one was removed from the feature list.”
Other details.
Line 360 has been fixed.
We thank the reviewer for their consideration. We clarified the importance of the investigation, the novelty of the work, and streamlined the manuscript by removing unnecessary text. We have done a thorough editing pass of the paper the fix language issues. We appreciate the opportunity to have made these extensive changes which have significantly improved the manuscript.
Reviewer 3 Report
The proposed paper deals with a machine learning model classifying smartphone activities through embedded IMU. Despite the topic is interesting, there are some relevant aspects which should be carefully addressed; accordingly, I suggest the following major revisions.
The final aim and the practical applications proposed by the authors are not so clear. The applications described (lines 43-45) don’t match with the results obtained in the paper.
The authors don’t describe thoroughly the experimental setup, especially the data collection is not well outlined. I deduced that each subject sequentially performs three conditions (cognitive load, sitting, walking) and four tasks for each condition (randomly proposed), but there isn’t a description supporting this deduction.
The numerousness of the sample (number of measures for each task relative to each condition) used to train the classifier is never mentioned, and the confusion matrices too, being normalized, do not allow for sample numerousness deduction. For this reason, the statement in the “Limitations” chapter, reporting the reduced number of subjects involved, indicates, in my opinion, a structural lack of the work more than a simple limitation. The number of subjects examined, though not explicitly mentioned, seems too small.
Furthermore, the implementation of a special app to present different contents (that are ready-made texts, videos, etc.) is far from a real context. Therefore, the poor results obtained by the authors would be even worst in the reality.
Finally, I suggest eliminating as a whole the cognitive load condition, not only because of the bad results obtained, but because it isn't a plausible protocol. In fact, it isn’t a real multitasking protocol but an alternation of listening and questions answering while performing the task.
The paper describes a rather complex aim that should be presented in a more organized way and realized according to a more rigorous scientific methodology.
The authors should add more bibliographic references, e.g. in lines 33-34 the statement should be supported by references. Figure 1, the axes names and the measurement units are missing Line 131, to validate the model’s performance the authors use the 5-fold cross-validation, but they should motivate the choice k = 5 Line 206, data is transmitted from the smartphone to the computer via wireless communication. Explain which type of communication is used and how often data is sent. Line 216-218, why did the authors choose the two sampling frequencies of 5 Hz and 50 Hz? Line 222, the data used for features' extraction is not pre-processed. The authors should motivate the choice not to apply any type of filtering considering the noise generally affecting the data acquired by inertial sensors. Line 323-325, the authors refer to tests carried out with windows of 10, 15, and 20 seconds and to the obtained accuracy. A table (as Table 3) shoud be added and described in detail. Table 3, the authors should explain the choice of 2.56, 5 and 8 second windows sizes to train the classification algorithms. In the text, the 2.56 s and 8 s windows are not mentioned at all. The caption of Table 3 doesn't specify the metric the percentages refer to. Line 360, the number of Table is missing. Table 8, the term "A few a times a day" must be replaced by "A few times a day". Lines 401-408, the model’s accuracies obtained at 50 Hz and 5 Hz are almost the same and this could be an added value for the study because at lower sampling rate the smartphone battery is saved. However, it is necessary to demonstrate this conclusion presenting the confusion matrix and the tables with the accuracy obtained for the 5 Hz solution. Furthermore, the results obtained at 50 Hz and 5 Hz must be compared with those already present in the literature and suggested by the authors in line 33 (without citing them).
Author Response
We thank the reviewer for saying that the topic of our paper was “interesting." We summarize each of the reviewer's comments as a subheading, and respond to each one in turn. We have made major changes to the manuscript, which further strengthened it.
Practical applications.
We outline two new arguments for this investigation in our introduction, and include an additional four citations for framing. We thank the reviewer for the reference on security as this is an important line of argumentation, along with personalization. We add the following to the introduction:
"Human activity tracking using smartphone sensors has been a key use case since smartphones have been released, in part due to smartphones' physical size, ubiquity, unobtrusiveness and ease of use. Whole-body human activities such as walking and running have been investigated but physically smaller human activities such as reading or typing have not been explored. Knowing whether a user is scrolling, typing or watching videos is valuable to better understand users. Digital content and experience providers benefit by tracking users' smartphone activities to inform personalization, as demonstrated by several recent patents ~\cite{hauser2011digital, linden2011content, smith2005content}. For example, news and social media applications log when users watch videos or read in order to present personalized content that is more likely to be engaging to each user \cite{thurman2011making}. If a news or social media app uses inertial measurement unit (IMU) sensors to learn that a user watches videos or reads when using other apps on their smartphone, they can present the user with more personalized content (i.e., more videos or more text, depending on what the user uses and likes).
The other side of this investigation is a question of security. It is not currently possible to access user action logs directly, due to smartphone data access privileges. Detailed logs of all user smartphone activities may be considered private or sensitive. As of 2019, smartphones do not ask users for permission to access IMU sensor data. A phone would need to be "rooted", in other words, an application would need to gain root access control, in order to access activity data directly. This may be because current IMU applications do not seem as privacy-sensitive as other permission-protected sensors such as Global Positioning System (GPS), the microphone and camera. However there is growing evidence that IMU sensors can be used for nefarious purposes, such as location tracking \cite{narain2016inferring} and password sniffing~\cite{ning2018deepmag}."
Experimental setup.
We have thoroughly edited the experimental setup section to clarify the order of the tasks.
Numerousness of sample.
We state the number of samples used for model training as follows, and further clarify the number of samples with additional detail:
“In total of 207 blocks of 2-minute activity data were recorded from the 21 participants. 3 blocks were invalid due to hardware failure.”
“With the 5-second window size, 4639 samples of data was extracted from the 50 Hz data, and 3620 samples form the 5 Hz data.”
In addition, we used accuracy vs sample size plot to evaluate how sample sizes can affect the model performance:
“To explore whether the data we collected was sufficient to train the model, an accuracy vs sample size plot was created for each of the three model, as shown in \ref{fig:accPlot}. The plots suggests that the accuracy of the model became fairly stable when the sample size was greater than 1500. This supports that the amount of data we collected was sufficient to produce a valid result.
\begin{figure}[h]
\centering
\includegraphics[width=\textwidth]{images/accPlot.PNG}
\caption{Accuracy vs sample size plot for the three models}
\label{fig:accPlot}
\end{figure}
”
Implications of results.
The reviewer writes: “the implementation of a special app to present different contents (that are ready-made texts, videos, etc.) is far from a real context. Therefore, the poor results obtained by the authors would be even worst in the reality.”
We agree with the reviewer’s prediction, and we explain why this is not a flaw of our research. We wanted to investigate whether human smartphone activity recognition is possible to accurately recognize by IMU sensors. We obtained a result that was not very high, which still has strong implications for personalization and privacy, because we learned that if apps use sensors to try to deduce what users are doing on their smartphones, they cannot do this very accurately. Furthermore, IMU sensors may not represent a security concern for detection of smartphone activities because the movement traces are not distinct enough for them to be accurately detected. We feel like this lack of accuracy is important for the research community to know, and therefore not a flaw.
Eliminating cognitive load condition.
We agree that this part of the investigation was less useful. We have removed all mentions of cognitive load, reducing our manuscript from 20 to 18 pages. Instead of including that text in this manuscript, we cite theses that provide those details.
We address the reviewers specific comments below.
The authors should add more bibliographic references, e.g. in lines 33-34 the statement should be supported by references.We have clarified that no other articles have investigated the recognition of human smartphone activity from IMU sensors.
“The area of human activity recognition has focused on recognizing physically larger human activities (walking, etc.) \cite{hassan2018robust, ronao2016human, voicu2019human,bayat2014study, ustev2013user, kwapisz2011activity, anguita2013public, khan2010human, shoaib2016complex, lu2017towards} but to date none have tried to recognize what the person is doing on the smartphone itself, namely reading, typing, scrolling or watching.”
Figure 1 has been edited and measurement units and axes names have been added.
Line 131, to validate the model’s performance the authors use the 5-fold cross-validation, but they should motivate the choice k = 5
We point out the area were we explain this choice:
“We used the k-fold cross-validation technique to evaluate the performance of the algorithms. To minimize the bias between participants, no participant's data appears in two folds. Therefore, 5-fold cross-validation was used because the data we collected from the participants was not enough to perform the 10-fold cross-validation (9 participants' data were collected in 5 Hz). The data used for training the model is divided into five sets, where each set consists of data from two participants on average. Among the five sets, one of them is selected to be the test set, and the remaining four sets are used to train the classifier, and the accuracy is evaluated on the test set. This process is iterated for each of the sets, and the final validation result is the average of all iterations. We aimed to classify the human smartphone activities into 10 labels, including sitting scrolling, sitting reading, sitting typing, sitting watching, sitting idle, walking scrolling, walking reading, walking typing, walking watching, and walking idle. ”
Line 206, data is transmitted from the smartphone to the computer via wireless communication. Explain which type of communication is used and how often data is sent.
The wireless communication we used was Bluetooth, which we mention and explain in the “participant recruitment and data acquisition” section, however we are happy to also specify in this section.
Line 216-218, why did the authors choose the two sampling frequencies of 5 Hz and 50 Hz?
We explain:
“Sampling frequencies can have an impact on the performance of the classification. And, power consumption differs across sampling frequencies. To examine the effect of different sampling frequencies to the model performance, we tested the two sampling frequencies that the smartphone can provide, which are 5 Hz and 50 Hz. In our study, 9 participants’ data was sampled with a frequency of 5 Hz, and the remaining 12 participants were sampled with a frequency of 50Hz..”
Line 222, the data used for features' extraction is not pre-processed. The authors should motivate the choice not to apply any type of filtering considering the noise generally affecting the data acquired by inertial sensors.
We add a reference to support our choice to use raw data for feature extraction:
“Mourcou et al. compared Apple and Samsung Galaxy smartphone IMU sensors to a gold standard industrial robotic arm IMU sensors and found them to be comparable \cite{mourcou2015performance}. Thus we used the raw data directly for feature extraction without any pre-processing filters.”
We added table to show the cross-validation result for these window sizes.
We explain:
“Based on previous ranges of suitable window sizes for similar investigations \cite{hassan2018robust, ronao2016human, voicu2019human, ustev2013user, anguita2013public, shoaib2016complex}, we systematically explored sizes of 2.56 seconds, 5 seconds, to 8 seconds.”
The caption of Table 3 has been fixed by clarifying it is the cross validation score and its standard deviation.
Line 360 has been fixed.
Typos in Table 8 have been fixed.
Lines 401-408, the model’s accuracies obtained at 50 Hz and 5 Hz are almost the same and this could be an added value for the study because at lower sampling rate the smartphone battery is saved. However, it is necessary to demonstrate this conclusion presenting the confusion matrix and the tables with the accuracy obtained for the 5 Hz solution.
We agree this will be helpful for the reader. We add an additional confusion matrix and the following text:
“With lower sampling frequency, the model would have difficulty in classifying activities between scrolling and typing, especially when the user is walking. Therefore, to achieve higher accuracy, higher sampling rate is necessary, which means it is not feasible to maintain a reasonable accuracy and reduce power consumption by using lower sampling frequency.”
Furthermore, the results obtained at 50 Hz and 5 Hz must be compared with those already present in the literature and suggested by the authors in line 33 (without citing them).
We edited line 33 to clarify that no prior studies have investigated the recognition of smartphone activities (typing, reading, scrolling, watching) using IMU sensors. Thus we cannot compare our results to others.
We thank the reviewer for their insights. We clarified the practical applications and experimental setup of the investigation, streamlined the manuscript by removing the text on cognitive load and fixed the issues generously pointed out by the reviewer. We have done a thorough editing pass of the paper the fix language issues. We appreciate the opportunity to have made these extensive changes which have significantly improved the manuscript.
Reviewer 4 Report
This paper presents an interesting project of classifying activities of using smart phones. It would be improved if more intensive experiments are performed with various types of users in terms of ages.
Author Response
We thank the reviewer for saying that our paper “paper presents an interesting project of classifying activities of using smartphones" and we thank the reviewer for the insight on strengthening the work with more diverse participants and add this as future work:
“To improve this research and make the results more realistic, more participants need to be recruited from different user groups (i.e., different occupations and different age groups).”
We have done a thorough editing pass of the paper the fix language issues. Please note that we have completed other clarifications and streamlining of the manuscript. Most notably, we highlight results regarding classification of smartphone activities by removing the discussion of cognitive load. We appreciate the opportunity to have made these extensive changes which have significantly improved the manuscript.
Round 2
Reviewer 2 Report
The authors have addressed all my concerns.
Author Response
We thank the reviewer for reviewing our paper for the second round, and saying "The authors have addressed all my concerns."
we have also made changes to correct typos in the paper located in line 104, 361 and in table 5 and 6. Also, we have done a thorough editing pass of the paper the fix language issues.
Reviewer 3 Report
The authors adequately complied to the requests made in my previous review. Nonetheless, the weaknesses relating to the reduced number of persons involved in the tests and the low performance achieved by the classifiers remain identical to the previous version, even if these have been justified by the authors. Despite these shortcomings, the work is sufficiently worthy of being published in this journal.
Finally, there are some typing errors:
- Line 104: “extensively” is repeated twice
- In Tables 5 and 6 the presented algorithms are all referred to the championship frequency of 50 Hz, conversely, I suppose that the last three are referred to the 5 Hz frequency
- Line 361: “nince” must be replaced with “nine”
Author Response
We thank the reviewer for reviewing our paper for the second round, and saying " the work is sufficiently worthy of being published in this journal."
The weakness addressed by the reviewer was not improved because no further experiment was carried out during the time of paper submission. However, the result can be improved in the future by involving more participants.
We have addressed all the typos been pointed out by the reviewer located in line 104, 361 and in table 5 and 6.
We have done a thorough editing pass of the paper and fixed language issues. We appreciate the opportunity to have made these extensive changes which have significantly improved the manuscript.